# Energy and biomass distribution in soil food webs of temperate and tropical forests

Anton M. Potapov [1,2,3,4] ✉, Irina Semenyuk[5,6], Sarah L. Bluhm [1], Valentyna Krashevska[1,7], Alexey Kudrin[8], Varvara Migunova[5], Melanie M. Pollierer [1,9], Oksana Rozanova [5], Sergey M. Tsurikov [5], Zheng Zhou [1], Andrey G. Zuev [2,5], Anna I. Zueva[2,5], Stefan Scheu [1,10,11] & Alexei V. Tiunov [5,11]

Soil food webs channel most of the energy in terrestrial ecosystems. Temperate and tropical forests host different soil invertebrate communities, but consequent differences in the structure and functioning of soil food webs between these major biomes remain unknown. Here, we calculate energy fluxes to explore generic patterns in biomass and energy distribution across micro-, meso- and macrofauna in forest ecosystems spanning from southern taiga to tropical rainforests. Tropical soil food webs have either larger (monsoon forest) or smaller (rainforest) animal biomass than temperate ones, but have consistently higher energy flux, higher share of large organisms (macrofauna) in total biomass and distinct energy distribution. Specifically, tropical soil food webs have proportionally higher predation rates than temperate soil food webs and rely more on plant consumption (living roots), but less on bacterial, fungal and litter consumption. Earthworms act as food-web engineers promoting detrital energy pathways (litter, soil and deadwood consumption) in mixed broadleaved forests overriding climate-associated differences among forests. Our study shows a major change in soil food web from "brown" temperate to "green" tropical functional state, explaining functional implications of soil invertebrate community turnover across biomes.

Energy flux in the ecosystem, i.e., the total amount of energy transferred in food webs, is associated with both ecosystem biodiversity and functioning[1]. In terrestrial ecosystems soils host most of animal biomass[2] and diversity[3], and channel most of the energy through food webs[4]. In forest ecosystems 25–80% of the annual litterfall may be consumed by soil animals[5]. Still, many aspects of soil food-web structure and functioning remain enigmatic[6,7] and large-scale variations in total energy flux and energy channels originating from different basal resources[8] across climatic regions remain unknown.

The most comprehensive assessment of soil animal communities and related energy balances comes from the International Biological Program and dates back to 1982[9]. This review reported variations in

[1]J.F. Blumenbach Institute of Zoology and Anthropology, University of Göttingen, Göttingen, Germany. [2]Senckenberg Museum of Natural History Görlitz, Görlitz, Germany. [3]German Centre for Integrative Biodiversity Research (iDiv) Halle-Jena-Leipzig, Leipzig, Germany. [4]International Institute Zittau, TUD Dresden University of Technology, Zittau, Germany. [5]A.N. Severtsov Institute of Ecology and Evolution, Russian Academy of Sciences, Moscow, Russia. [6]Joint Russian-Vietnamese Tropical Center, Ho Chi Minh City, Vietnam. [7]Functional Environmental Genomics, Senckenberg Biodiversity and Climate Research Centre, Frankfurt, Germany. [8]Institute of Biology, Komi Scientific Centre, Ural Branch, Russian Academy of Sciences, Syktyvkar, Russia. [9]Institute for Forest Protection, Julius Kühn-Institute (JKI) - Federal Research Center for Cultivated Plants, Quedlinburg, Germany. [10]Centre of Biodiversity and Sustainable Land Use, University of Göttingen, Göttingen, Germany. [11]These authors contributed equally: Stefan Scheu, Alexei V. Tiunov. ✉e-mail: potapov.msu@gmail.com

soil animal biomasses, community composition and metabolism across different biomes, concluding that soil animals contribute on average only about 5% to the total soil respiration[9]. However, more recent studies highlighted the importance of soil animals in the terrestrial biomass pool[10] and showed that consumer communities directly and indirectly can change respiration and decomposition rates by tens of percentages, especially in tropical ecosystems[11–13]. Over the last years, metabolic theory of ecology[14] and energy flux approaches[1,15,16] provided new tools to uncover functional roles of consumers in ecosystems, but they are yet to be used to understand large-scale variations in the structure and functioning of soil food webs.

Soil animals are traditionally divided into microfauna (<0.2 mm in body diameter, mainly nematodes) living in water films, mesofauna (0.2–2 mm, mainly springtails, mites and enchytraeids), and macrofauna (above 2 mm, most insects, myriapods, isopods, large arachnids and earthworms)[17]. The structure and functioning of soil food webs largely rely on the dominance of different animal size classes and taxonomic groups[16,18]. However, the changes in energy flux across size classes remains to be explored across climate types.

Basal resources, such as living plants, leaf litter, soil organic matter, bacteria and fungi, were recognized to structure soil food webs into energy channels, i.e., clusters of consumers that are trophically linked to corresponding basal resources[8]. Quantification of "trophic functions" associated with different energy channels, i.e., herbivory, algivory, bacterivory, fungivory, litter, deadwood, and soil consumption, and predation can inform on soil food web multifunctionality[16,19]. So far, the energy flux approach was not applied to test how major trophic functions in soil food webs change across large-scale environmental gradients, e.g., between tropical and temperate forests. Due to high temperatures in tropical ecosystems, the metabolic rate (i.e., the energy needed to sustain functioning of the organism) of organisms is high, necessitating high consumption rates[20]. Ecosystem production increases with temperature at a lower rate than metabolic losses of consumers[1], leading to energy limitation, high predation rates[2], competition for resources and prey among soil animals, and potential changes in the use of basal resources and related functioning in soil food webs of tropical forests. Testing these theoretical predictions would allow us to generalize food-web approaches across ecosystems to predict soil organic matter dynamics and better understand the factors driving terrestrial biodiversity.

Here, by reconstructing soil food webs and quantifying energy fluxes across temperate (southern taiga, mixed broadleaved, beech) and tropical forests (monsoon, rainforests), we aim at quantifying the variation in the structure and functioning of soil food webs along major environmental gradients. To achieve this, we collect data on density and body size of soil nematodes, microarthropods and macrofauna from 32 sites across four regions—Germany, European Russia, Indonesia and Vietnam in the periods of high activity of soil invertebrates (late summer or autumn in the temperate climate, early wet season in the tropical climate). We calculate individual body masses and metabolic rates[21] and use them together with stable isotope composition and other traits to reconstruct soil food webs[16] and calculate energy fluxes assuming a steady-state energetic system (i.e., losses = gains)[22]. Using calculated trophic functions we test the following hypotheses: (1) Tropical soil invertebrate communities have lower total biomass, but larger energy use per unit of time than temperate ones because of higher resource limitation and predation under high temperatures. (2) Proportionally more energy is processed by small-sized animals in the tropics because temperature poses limitations on body growth and more energy is invested in respiration. (3) Proportionally more energy is propagating to higher trophic levels in the tropics due to shallow litter layer and high animal activity resulting in lower search time and higher predator-prey encounter rate (i.e., predators spend less energy for prey searching and capturing). (4) Due

to high temperature, rapid decomposition (shallow litter layer), high energy demand and therefore high competition for resources in the tropics, consumption of easily accessible resources with fast turnover is promoted. Specifically, more energy is flowing via the bacterial, algal and plant channels in tropical ecosystems and via the detritus-fungal channel in temperate forests[23]. To uncover mechanisms behind these functional shifts, we also explore correlations of soil food web energy fluxes with key environmental variables such as temperature, precipitation, net primary production (NPP), soil pH, and litter C/N ratio.

## Results

### Animal biomass

Soil animal biomass (i.e., the sum of individual masses in a community) was at a maximum in monsoon forests ($50.4 \pm 25.1$ [1 SD] g fresh biomass m$^{-2}$) and at a minimum in rainforest ($6.6 \pm 3.1$ g m$^{-2}$) and southern taiga ($7.8 \pm 3.3$ g m$^{-2}$). Mixed broadleaved and beech forests occupied intermediate positions (Fig. 1 and Table 1; Forest type effect $\chi^2_4 = 41.5$, $p < 0.0001$). The animal biomass was distributed evenly across litter and top 0–5 cm of soil except in mixed broadleaved forests, where biomass in litter was low (Fig. 1 and Table 1; Forest type x Layer interaction $\chi^2_4 = 12.9$, $p = 0.0117$). Dominant taxonomic groups were present across all forest types with the exception of termites (present only in the tropics) and diplurans (absent in southern taiga). Earthworms represented a major part of the biomass in all forests, followed by beetles, spiders, and oribatid mites. Overall, biomass composition shifted from nematodes and microarthropods to macrofauna towards the tropics. Earthworms represented more than 90% of the total biomass in mixed broadleaved forests (Fig. 1).

### Body mass spectrum

There was a gradual increase in the biomass – individual body mass slope (i.e., increase in large-sized animals) from the taiga to the tropics, with the peak in monsoon forests (Fig. 2a; Size class effect $\chi^2_4 = 112.8$, $p < 0.0001$; Size class × Forest type interaction $\chi^2_{16} = 57.1$, $p < 0.0001$). No significant difference between soil and litter in the slope was observed (Size class x Layer interaction, $p = 0.9612$). The energy flux−body mass slope was slightly positive and similar in monsoon, taiga, and mixed broadleaved forest (i.e., more energy is flowing to large-sized animals), being slightly negative in rainforest and beech forests (Size class effect $\chi^2_4 = 62.1$, $p < 0.0001$; Size class × Forest type interaction $\chi^2_{16} = 57.7$, $p < 0.0001$). However, the body mass−biomass spectrum was remarkably non-linear with especially low biomass of the "small macrofauna" body mass class (50 μg to 1.6 mg) in taiga and mixed broadleaved forests. Generally, across forest types biomass was highest in the "large microfauna and microarthropods" (1.6–50 μg) and "macrofauna" body mass classes (1.6–50 mg). This also applied for the energy flux−body mass spectrum, except in southern taiga, where "large microfauna and microarthropods" dominated (Fig. 2b).

### Food webs and energy fluxes

Reconstructed soil animal trophic networks had more trophic levels and allocated more energy to macrofauna in the tropics than in temperate forests (networks are shown in Fig. 3 for illustration of the methodology and visual transparency of our results). The maximum trophic level was on average 4.2–4.4 in southern taiga, mixed broadleaved and beech forests, with ants, diplurans and spiders as top predators, and 4.5–4.6 in monsoon forests and rainforests, with diplurans and schizomids as top predators. The total energy flux was higher in monsoon forests ($1090 \pm 598$ mW m$^{-2}$) and rainforests ($227 \pm 62$ mW m$^{-2}$) than in beech ($109 \pm 48$ mW m$^{-2}$), mixed broadleaved ($32 \pm 13$ mW m$^{-2}$) and southern taiga forests ($26 \pm 11$ mW m$^{-2}$).

In terms of absolute values, trends in the trophic functions, i.e., predation, algivory, herbivory, bacterivory, fungivory, and litter, deadwood and soil consumption, generally followed the trend for the total energy flux (Supplementary Fig. 5). However, the relative

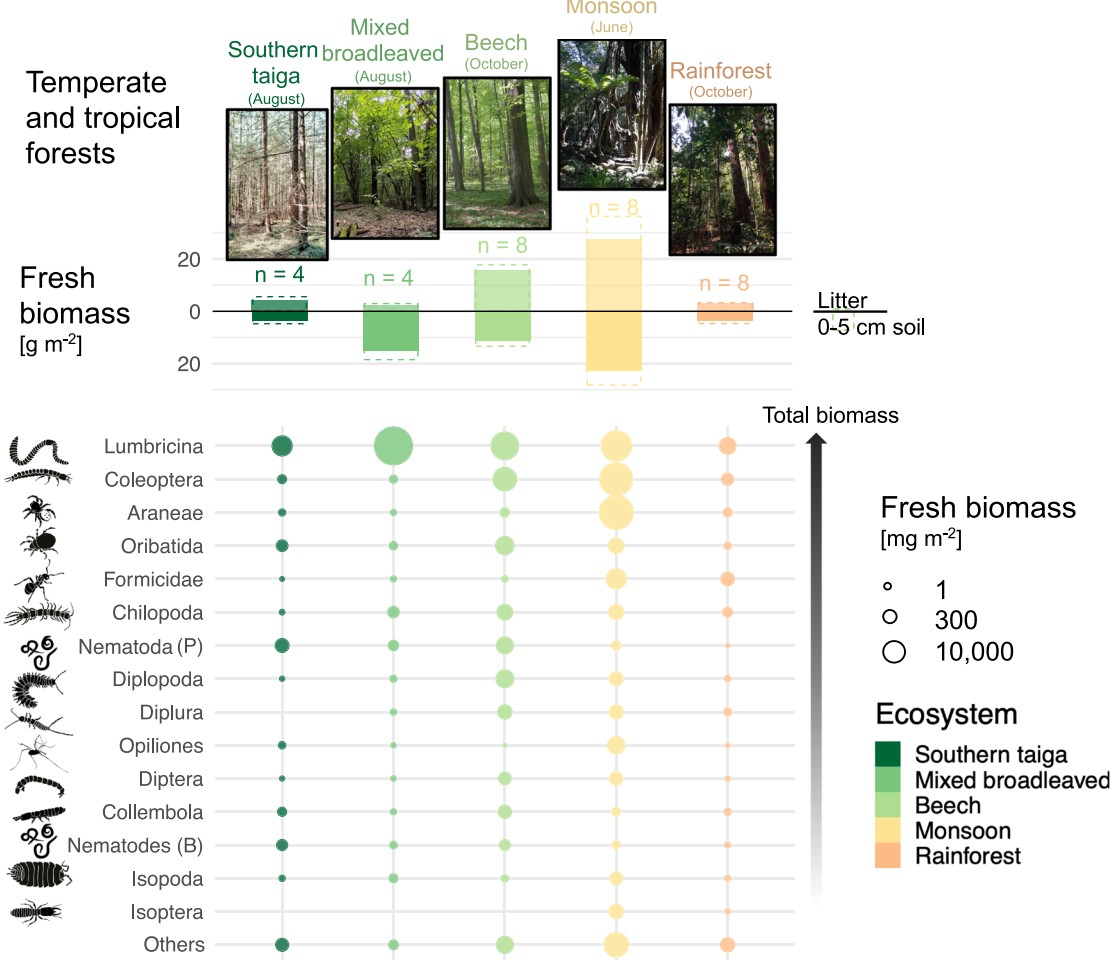

**Fig. 1 | Biomass of animal taxa in forests of different climatic regions.** The barplot shows the distribution of fresh biomass between litter and topsoil (0–5 cm; means ± 1 SD) in southern taiga, mixed broadleaved, beech, monsoon, and rainforests. The bubble plot shows the distribution of fresh biomass among dominant taxonomic groups and trophic guilds (for predatory and bacterivorous nematodes) in different forest types. The size of the bubbles is proportional to the biomass; animal groups are sorted by descending total biomass across the studied forests.

**Table 1 | Fresh biomass of soil animals in litter and top 5 centimeters of soil in temperate and tropical forests**

|  | Southern taiga (n = 4) | Mixed broad-leaved (n = 4) | Beech (n = 8) | Monsoon (n = 8) | Rainforest (n = 8) |
|---|---|---|---|---|---|
| Litter | 4.1 ± 3.1 | 2.1 ± 1.7 | 15.7 ± 5.9 | 27.5 ± 25 | 2.9 ± 1.1 |
| Soil | 3.7 ± 2.1 | 14.9 ± 7.1 | 11.2 ± 6.1 | 22.9 ± 15.1 | 3.7 ± 2.9 |
| Total | 7.8 ± 3.3 | 17.1 ± 8.7 | 26.9 ± 11.5 | 50.4 ± 25.1 | 6.6 ± 3.1 |

Units are g m$^{-2}$; means ± 1 SD are shown

contribution of these trophic functions to the total energy flux within each forest type differed considerably (Fig. 4). Soil food webs in monsoon forests and rainforests had proportionally higher predation rates than in beech, mixed broadleaved and southern taiga forests (29 vs 17–25% of the total energy flux) and relied more on plant consumption (19–24 vs 6–10%), but less on bacterial (6–7 vs 10–17%) and litter consumption (13–14 vs 18–32%). Algivory, representing mainly feeding on microalgae and lichens, varied between 0.5 and 3%, with highest values in monsoon forests. Fungivory was higher in southern taiga and beech forests (30–31%) than in other forest types (17–24%). Fungivory-to-bacterivory ratio was low in southern taiga and mixed broadleaved forests (1.8 ± 0.2 and 1.3 ± 0.2, respectively) in comparison to beech, monsoon forests and rainforests (3.1 ± 0.8, 3.0 ± 0.8 and 3.4 ± 1.3, respectively). Wood and soil feeding was higher in tropical

forests (2–4% each) than in southern taiga and beech forests (1–2% each). In mixed broadleaved forests the detrital energy pathways dominated (litter, soil, and deadwood consumption jointly comprising 44% of the total energy flux), while predation was low (17%).

Different animal groups contributed the most to each trophic function in different forest types (Fig. 4). Across forests, nematode groups accounted for most of bacterivory (61–87% of the total bacterivory), fungivory (19–60%; except mixed broadleaved forests) and herbivory (21–67%; except monsoon forests), and dominated predation in temperate forests (9–67% of the total predation). Macrofauna groups dominated predation in tropical forests and herbivory in monsoon forests. Litter consumption in rainforest, southern taiga, and beech forests was contributed mainly by springtails (31–44%), while millipedes (20%) and earthworms (82%) dominated this function in monsoon and mixed broadleaved forests, respectively. Earthworms performed most of the soil consumption across forests (48–96%) and most of deadwood consumption in temperate forests (35–91%). The latter function was mainly contributed by termites in monsoon forests (41%, although their contribution to the litter and deadwood consumption is probably underestimated) and beetles in rainforests (46%).

**Association with environmental factors**

The five studied forest types were distinct in the energy flux distribution among taxonomic groups/trophic guilds (anosim $R = 0.87$;

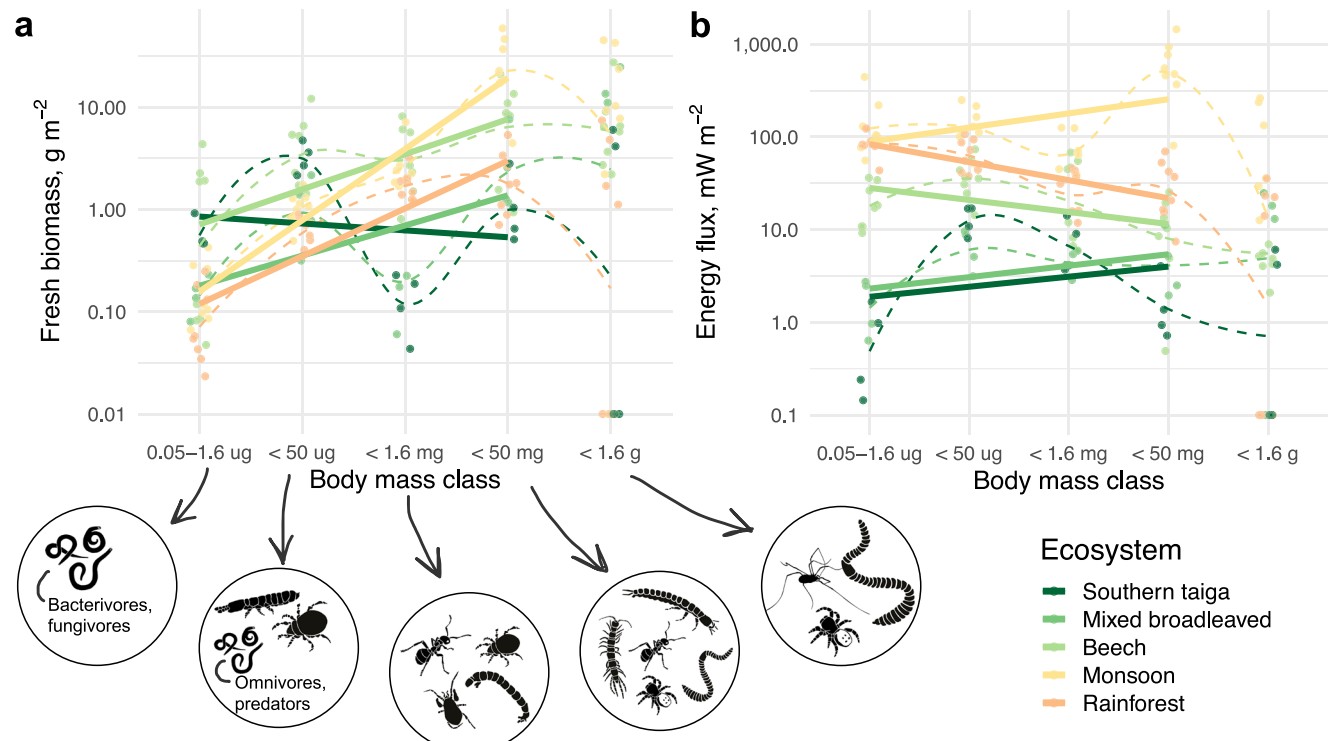

**Fig. 2 | Soil animal biomass and energy flux in different body mass classes in forests of different climatic regions.** The body mass spectrum was divided into five body mass classes using linear bins on a fresh body mass logarithmic scale: (1) microfauna (0.05–1.6 µg), (2) large microfauna and microarthropods (1.6–50 µg), (3) small macrofauna (0.05–1.6 mg), (4) macrofauna (1.6–50 mg) and (5) large macrofauna (0.05–1.6 g). Biomass (**a**) and energy flux (**b**) within each class were summed up per site (represented by points). Linear trends across classes are shown with solid lines (large macrofauna was excluded due to potential undersampling), non-linear smoothings are shown with dashed lines. Animal groups representing a major part of the animal biomass (>6–7%) in the respective class are shown with silhouettes.

$p = 0.001$; Fig. 5a). This distinction correlated with higher temperature, precipitation and NPP in tropical forests ($R^2 > 0.71$, $p < 0.001$; Fig. 5b). Higher percentages of herbivory and predation correlated with temperature, precipitation and NPP, being high in tropical, maily monsoon forests (energetic contribution of butterfly larvae, orthopterans and hemipterans). Higher percentages of bacterivory and fungivory (i.e., dominance of microbial energy channels) were positioned oppositely, being higher in temperate forests (energetic contribution of earthworms and nematodes; Fig. 5). Correlations of detritivory percentages with the energy distribution among taxa were not significant, but in general wood and soil consumption were associated with high temperature, precipitation and NPP, as opposed to litter consumption which was closely associated with high soil pH. Litter consumption and microbivory were associated with low litter C/N ratios, as opposed to algivory, predation, and herbivory (Fig. 5b).

**Sensitivity analysis**
Feeding preferences for bacteria, fungi, living plants and in many cases for different detritus pools used for food-web reconstruction were based on published data rather than direct measurements. Therefore, we tested how the degree of omnivory affects reported differences in trophic functions among forest types. Across the range of coefficients (i.e., preferences to additional food resources from 0 to 100% of the main resource), our sensitivity analysis confirmed higher percentages of predation and herbivory, and lower percentages of bacterivory and litter consumption in the tropics (Supplementary Fig. 6). The effects of the omnivory coefficient was the strongest for fungivory, litter and deadwood consumption in mixed broadleaved forests because these functions were mainly supported through auxiliary feeding of earthworms. Across all forest types and omnivory coefficients, the following allocation of energy in soil animal food webs was recorded: 12–41% of the total energy flux for predation, 1–4% for algivory, 3–23% for herbivory, 2–23% for bacterivory, 9–30% for fungivory, 9–60% for litter consumption, 0–7% for deadwood consumption and 1–12% for soil consumption.

## Discussion
By reconstructing soil food webs and estimating energy fluxes in multiple forest types in Germany, Russia, Vietnam, and Indonesia, we quantitatively analyzed the variation in structure and functioning of soil food webs across tropical and temperate forests. Our comparison is based on snapshot data of peak soil fauna activities, so the absolute values should be treated with caution. Nevertheless, the proportional changes of different trophic functions are likely representative for the climate types. In line with our first hypothesis, energy flux in soil invertebrate communities was larger in tropical than in temperate forests. However, tropical forests had both the largest (monsoon) and the smallest (rainforests) total biomass. In contrast to our second hypothesis, biomass and energy flux were concentrated in macrofauna, rather than in micro- and mesofauna size classes in the tropics. Finally, we showed substantial differences in trophic functions in soil food webs between temperate and tropical forests. In part supporting our third and fourth hypotheses, tropical soil food webs had proportionally higher predation and herbivory (fast energy channeling), and proportionally lower litter consumption (slow detritus-based energy channeling). Unexpectedly, we also observed a clear decrease in relative bacterial feeding despite the earlier documented relative increase in bacterial biomass toward the tropics[23].

Soil animal biomass in rainforest comprised only about 7 g m$^{-2}$ (fresh weight), being at the lower end of the values in temperate forests (8–27 g m$^{-2}$). This fits published figures that tropical forests have low soil animal biomass (1.8–3 g m$^{-2}$ dry weight)[9,10] in comparison to

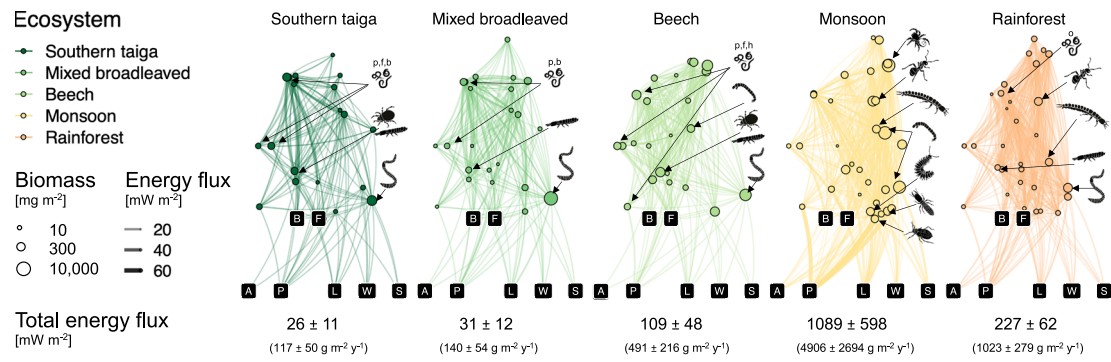

**Fig. 3 | Soil food webs in forests of different climatic regions.** Soil food-web topologies across nematodes, microarthropods and macrofauna depicted using flow charts with bubbles representing nodes (the size is proportional to the biomass) and connecting lines representing energy fluxes (the thickness and transparency is proportional to the energy flux). Black square nodes represent basal resources: A algae, P roots (or shoots) of living plants, B bacteria, F fungi, L litter, W deadwood, S soil organic matter. Animal nodes are ordered horizontally according to the body mass and vertically according to trophic level. Animal nodes channeling a major part of the animal energy flux (>5%) in the respective forest types are highlighted with silhouettes (Fig. 1; lowest silhouette in monsoon forest represents Hemiptera). Nematoda are divided into predators (p), fungivores (f), bacterivores (b), herbivores (h) and omnivores (o). The networks represent average biomasses and fluxes. Individual networks are given in the Supplementary Information. The total energy flux values are additionally converted into fresh biomass flux for comparability reasons.

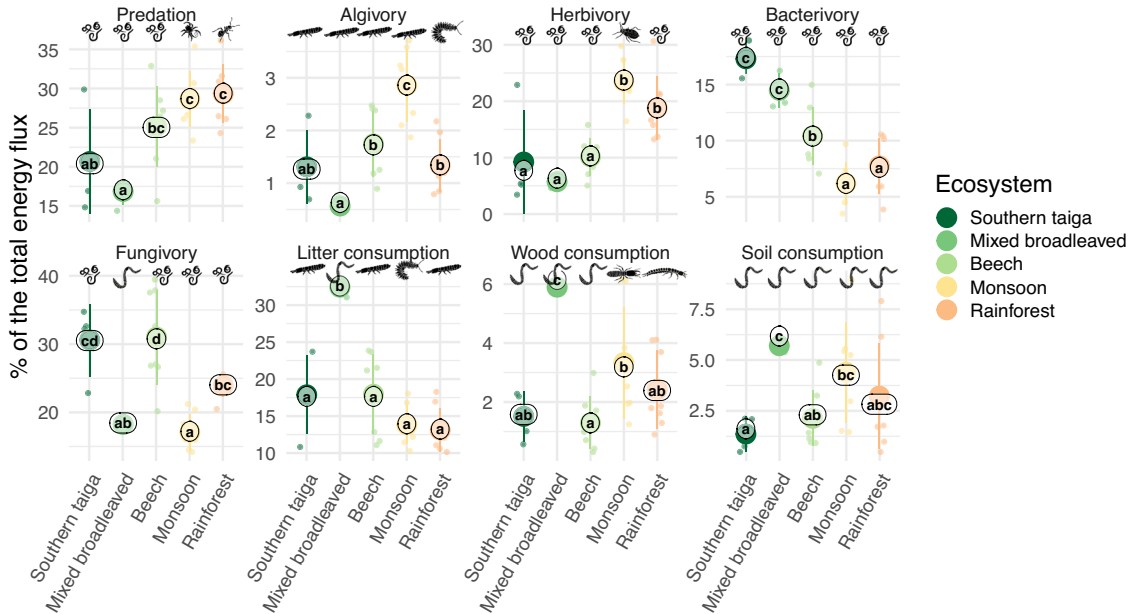

**Fig. 4 | Trophic functions in soil food webs in forests of different climatic regions.** Each trophic function represents the sum of all energy fluxes outgoing from a specific resource/prey: predation–animals, algivory–algae, herbivory– living plants, bacterivory–bacteria, fungivory–fungi, litter consumption– litter, wood consumption–dead wood, soil consumption–soil organic matter. Functions are scaled to the total energy flux (percentages). Arithmetic means ± 1 SD are shown with large points, each small point represents a site. Means sharing the same letter within resource types are not significantly different (Tukey's HSD test for beta regressions; labels show least-squares means from the beta regression models). The animal group contributing most to the respective function in the respective forest type is highlighted with silhouette (Fig. 1; Hemiptera are shown in herbivory). Differences in the absolute fluxes are given in Supplementary Fig. 5.

temperate coniferous and deciduous forests (2.4–8.0 g m$^{-2}$)[9], however, biomass in the latter is largely represented by earthworms[9], while arthropods do not have considerably higher biomass in temperate than in tropical forests[9,10]. We hypothesized that tropical soil food webs cannot sustain large community biomass because of higher metabolic rate and increased resource limitation and predation. Indeed, the community energy flux in rainforests was higher than in temperate forests despite considerably lower community biomass. Our energy flux estimates were only partly in line with existing figures on respiratory metabolism of soil fauna by International Biological Program[9]: 32–113 versus 12–94 mW m$^{-2}$ in temperate broadleaved forests (converted from 389 to 2968 kJ m$^{-2}$ y$^{-1}$), 26 versus 1–19 mW m$^{-2}$ in temperate coniferous forests and 236 versus 4–49 mW m$^{-2}$ in

rainforests, respectively. However, respiratory metabolism estimations calculated by Petersen and Luxton do not consider losses due to predation and assimilation inefficiency (a large part of consumed detritus is not assimilated), both considered in our energy flux calculations. As consumption is ca. 3 times higher than respiration in soil animal communities[9] and predation accounts for 17–29% of the entire energy flux (Fig. 4), the above-listed numbers align better, especially considering higher predation rates in the tropics.

Monsoon forests had by far the highest soil animal biomass (50 g m$^{-2}$ fresh weight) and energy flux (10 times higher energy flux than beech forests and 5 times higher than rainforests). Currently, monsoon forests are underrepresented in the literature with only few studies reporting biomass of specific taxa and covering only a limited

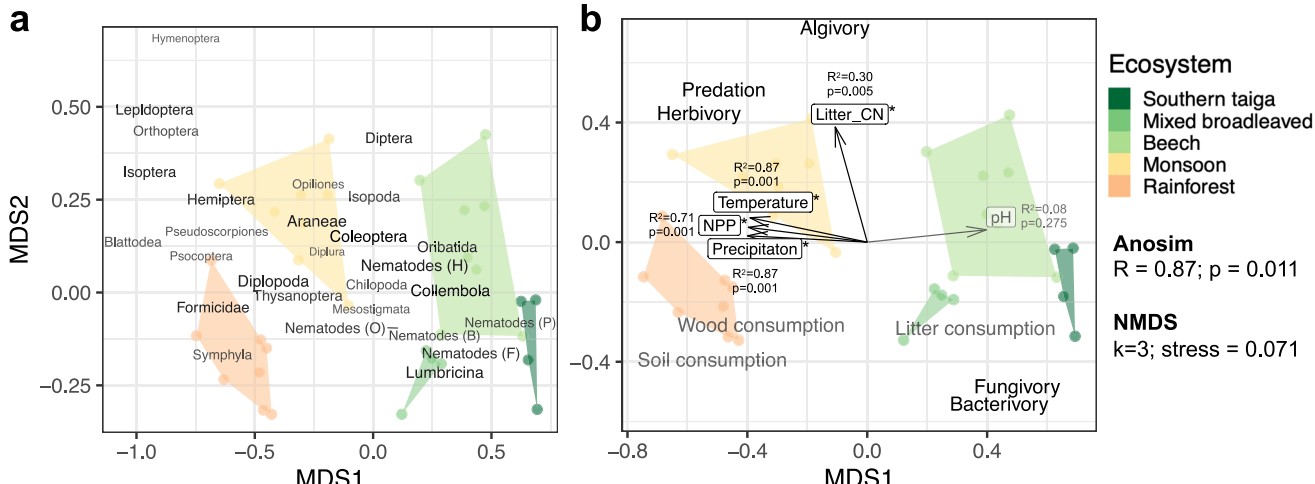

**Fig. 5 | Association of the energetic structure of soil animal communities with environmental variables and trophic functions across forest ecosystems.** The first two axes of non-metric multidimensional scaling (MDS) based on the absolute energy fluxes of the corresponding trophic groups/taxa are shown. Each point represents a site, polygons delineate forest types (shown with different colors). Trophic groups/taxa are positioned according to their contribution to MDS1 and 2 axes, groups with higher total energy fluxes across all sites are given bigger and brighter (**a**). Environmental variables and trophic functions (see Fig. 4) are positioned correspondingly to their correlations with the MDS axes; significant correlations with the scaling are given in black and marked with asterisks (ordinary least squares regression with two-sided permutation significance tests), non-significant in gray; for environmental variables, the strength of correlations are also given in text (**b**). NPP net primary production.

range of size classes[24,25]. The high energy flux in monsoon forests and rainforests is correlated with conditions promoting biological activity: high temperature, precipitation and NPP (Fig. 5). However, high energy flux in monsoon forests is unlikely to be explained by ecosystem productivity alone, since there are no such drastic differences in NPP among the studied ecosystems (global NPP projections averaged 4091, 3095, 4690, 6050 and 7177 g m² y⁻¹ of fresh biomass across our latitudinal gradient in the five forest types)[26,27]. One potential explanation for the difference is a stronger seasonal variation in precipitation in monsoon forests in comparison to equatorial rainforests. Our sampling was done at the beginning of the wet season, when soil invertebrates are especially abundant and active. It can be expected that annual energy flux in soil food webs is distributed unevenly across the year in monsoon forests, with resource accumulation in the dry season and resource consumption in the wet season. To estimate this bias, we used data on the seasonal variation of macrofauna populations in the area of our sampling sites[28] and found that annual average biomass of macrofauna represents approximately 90% of the June biomass, suggesting that this bias cannot explain the order-of-magnitude difference in energy fluxes between monsoon and temperate forests. The strong difference to temperate soil food webs, which also have strong seasonality, is likely to be due to the higher temperature, precipitation and NPP in the tropics. Comparison of the NPP values with the total energy fluxes suggests that soil animals consume 3–5% of the NPP in southern taiga and mixed broadleaved forests, 10% in beech forests, 14% in rainforests, and 81% in the monsoon forest (wet season). These estimates should be treated with great caution since (1) strong seasonality may overestimate the total consumption by soil animals in monsoon forest and (2) metabolic rate coefficients originate from laboratory experiments and may result in biassed absolute energy fluxes. Despite consumption of 81% of NPP looks unrealistic, personal observations of the authors in the field sites in Vietnam suggest that nearly all leaf litter accumulated during the dry season in which litterfall occurs disappears within a few weeks or even days after the first rains due to high feeding activity of termites and other large detritivores. This aligns with recently published estimates of litter consumption by soil fauna[5]. Approximately 80% of consumed litter returns to the system in the form of unassimilated material (feces; see "Methods") and further enters the nutrient and energy cycles in the soil

and termite mounds. While keeping in mind the uncertainty of absolute values, we still can conclude that tropical animal soil food webs channel a much larger part of the NPP than temperate ones and this coincides with lower accumulation of topsoil organic matter in the former[29].

We further recorded changes in the biomass and energy distribution along body mass classes depending on the forest type. In general, macrofauna dominated in tropical forests, resulting in a more positive body mass–biomass slope and confirming previous taxonomy-based analyses[9,30]. This contrasts the general pattern that ectothermic organisms in general grow bigger in cool conditions[31]. Moreover, this also contrasts the positive relationship between phosphorus availability and the slope of the soil animal body mass spectrum, because mature tropical ecosystems are typically limited by phosphorus[32] (although we did not have direct measurements of phosphorus from all our sites). Potentially, counter-acting ecological mechanisms give advantage to large-bodied organisms, allowing them to accumulate higher community biomass in the tropics. We suggest several potential ecological explanations (cf., e.g., Chown and Gaston[33]): (1) generation time of soil animals in temperate ecosystems is associated with season length and they are unable to grow large due to the relatively short warm season. (2) There is intense competition for patchy-distributed resources in tropical forests, which makes higher mobility an advantage for foraging them, and therefore there is a selection for large body size (e.g., ants and termites are effective foragers that dominate in the tropics). (3) Tropical systems are energy limited and large animals have lower metabolic rates per biomass unit and a higher tolerance of starvation and desiccation. (4) Litter in the tropics is of low quality requiring large shredding mechanisms to mechanically process it. Notably, we did not find differences in the body mass–energy flux relationship between tropical and temperate ecosystems, contrasting observed differences in the body mass–biomass relationship (Fig. 2b). This "flattening" of the energetic slope in the tropics (in comparison to temperate forests or the biomass slope) is likely to be explained by high metabolic rates and predation pressure on small animals in comparison to large animals[34]. Both factors increase metabolic losses and thus energy flux in the lower part of the size spectrum. Overall, temperature- and nutrient-related mechanisms driving local/regional body mass distributions in soil communities[29,35]

may contrast patterns at larger scales, which calls for further cross-scale studies.

In most forests, we observed non-linear body mass spectra with a double mode distribution. Specifically, biomass and energy concentrated in the "large microfauna and microarthropods" and "macrofauna" body mass classes, whereas "microfauna" and "small macrofauna" had low biomass and energy flux in comparison to other body mass classes. This rather general pattern resonates with the concept of size compartmentalization in soil food webs, implying that different size classes function as rather independent energetic systems[34]. The reasons for relatively low biomass and energy flux in the "small macrofauna" size class (50 µg to 1.6 mg) remain obscure and might point to some disadvantages of having this body size in soil systems. For example, this body size might be too large to selectively feed on microorganisms and use soil pores as a shelter, but too small to effectively shred leaf litter and create pore space. An alternative explanation is the dynamic nature of the size spectrum due to the soil insect emergence and other phenological rhythms. The latter, however, is unlikely to explain the similarities among forest types, as they were sampled across seasons.

Changes in the total energy flux across forest types were paralleled by community turnover and re-distribution of energy along different trophic levels and energy channels based on different resources. Aligning with our expectation, predation accounted for a larger part of the total energy flux in the tropics in comparison to temperate forests, confirming the latitudinal pattern observed in aboveground arthropods[36]. This is also in line with experiments reporting increased predation at warmer conditions[37,38]. We suggest that high predation in tropical forests is connected to a higher encounter rate due to higher activity at high temperatures and limited shelter space due to faster litter disappearance. We also observed a strong association between predation and herbivory, suggesting that predation in soil food webs is tightly associated with the "green" energy channel and offering good promises for biological pest control (although this pattern is yet to be tested in agroecosystems).

While soil food webs in temperate forests relied mainly on litter and microorganisms (fungi and/or bacteria) as basal resources, rainforests and monsoon forests relied more on living plants, deadwood, and soil organic matter (the latter two resources were also important in mixed broadleaved forests; Figs. 4 and 5). These differences in resource consumption probably illustrate different patterns or carbon flows at the ecosystem level. Due to environmental limitations on decomposition rate, litter accumulates in many temperate forests and hosts large microbial biomass stocks, providing also living space and resources for soil animals[39]. By contrast, in tropical forests litter is consumed and decomposed rapidly[39,40], which probably forces soil animals to more intensively feed on living plant tissues or on recalcitrant resources, such as deadwood or soil organic matter. High herbivory in soil food webs supports the hypothesis that biogeochemical cycles in tropical ecosystems are shortened with consumers in soil being forced to feed on plant biomass before it is converted to detritus. Exaggerating the observed differences, we propose two contrasting soil food web functional states, the "green" state, relying more on freshly-fixed plant biomass with intensive predation (dominated mainly by insects), and the "brown" state, relying more on old accumulated detritus and less active transfers to the high trophic levels (dominated by earthworms or micro- and macroarthropods). These states can be assigned to ecosystems, but also to ecosystem stages: as suggested by Odum, ecosystem development starts from early systems with high turnover and low biomass-to-energy ratio relying on green plants to mature slow systems with high biomass-to-energy ratio relying on detritus[41]. The relative contribution of living and dead plants to soil food webs across seasons remains to be tested.

We observed unexpectedly low bacterivory and high fungivory-to-bacterivory energy flux ratio in tropical soil food webs, despite low fungi-to-bacteria ratios typically observed in tropical soils[23]. In part, this may be due to spatial separation of bacteria and soil fauna in the tropics. Due to shallow litter in the tropics, bacterial biomass is concentrated in soil and is only accessible to a limited number of soil-feeding macrofauna (earthworms, termites) and some micro- and mesofauna. By contrast, fungi form macroscopic structures on the soil surface and form extensive mycelium to access the recalcitrant litter. While our study did not consider protists as a major group of bacterial feeders, currently we lack robust data to compare protist biomass and consumption rates across global forests. Although the estimated bacterivory change requires validation using state-of-the-art empirical methods[42], the discrepancy between microbial biomass stock and related energy fluxes in soil food webs is remarkable and emphasizes the necessity for food-web approaches to understand ecosystem functioning and stability.

Different soil animal taxa have different roles in soil food webs and therefore contribute differently to ecosystem functioning[43]. Here, we highlight several important cross-ecosystem patterns observed in our study. As "ecosystem engineers"[44] and "ecosystemivores"[45], earthworms change the soil system and this applies also to soil food webs[46]. In mixed broadleaved forests distinct energy distribution associated with high earthworm biomass is a good illustration of these effects: detrital energy pathways dominated (litter, soil, and deadwood), while predation and fungivory were reduced in comparison to southern taiga and beech forests. Negative effects of earthworm abundance on predation[47] and arthropod abundance[48] were reported before. Therefore, in addition to already proposed titles, earthworms can also be called 'food-web engineers'.

Among mesofauna, we showed contrasting patterns in springtails and oribatid mites. Across forests, oribatid mites had higher biomass than springtails (Fig. 1), however, springtails contributed more to energy flux and even dominated in some functions such as litter consumption (Fig. 4). This biomass-energy flux discrepancy is explained by two factors: (1) springtails have higher metabolic rates per mass unit than oribatid mites, whose metabolism is relatively low[21]; (2) oribatid mites are well protected[49] and thus suffer much less energetic losses (i.e., need less energy to sustain their populations), which is included in our food-web reconstruction. It has been shown recently that springtails and mites represent ca. 95% of the total arthropod abundance on Earth, with mites representing two-thirds of it[10]. Here, we showed that the contribution of springtails and mites to energy flux is similar across forest systems in different climates, or even larger for springtails.

Finally, nematodes played a very important role in herbivory, fungivory, and especially bacterivory. These small worm-like invertebrates occupy all key positions in micro-food webs[50], having even the largest share in predation in temperate forests (Fig. 4). To the best of our knowledge, our study provides the first comparison of energy flux in micro- meso- and macrofauna across different forest types and latitudes, showing that contributions of these size classes are similar and the dominant size class depends on specific forest type. However, our soil food-web reconstruction ignored protists as another major consumer group in micro-food webs[51]. Therefore, our study quantitatively supports the view that the micro-food web is the energetic core of the soil food web responsible in large for nutrient cycling, while the macro-food web ensures transformation of organic matter through detritivory. At the same time, we highlight that the roles of different groups and body size compartments change with forest types.

Despite rapidly developing methods and accumulated knowledge on soil food-web structure, some aspects of our study are based on assumptions. Here, we highlight potential biases to point out remaining knowledge gaps. Our study represents a snapshot of dynamic food webs. While we selected mature forests in every region, they are expected to show seasonal dynamics[52]. Following climatic variations, these dynamics are higher in temperate and monsoon forests than in rainforests. Since we assessed soil animal communities in periods of

high activity, the average annual energy flux in rainforest may be underestimated in comparison to other forest types. Although we were not aiming at global-scale analysis due to a limited number of regions included, we believe that our results may well represent the latitudinal gradient because the differences in abundance and diversity of soil animals between temperate and tropical are larger than within them[9]. Although we did an extensive sampling at each of the study sites (6–42 spatially distinct samples collected per site), we admit that we unlikely captured the entire diversity and complexity of each forest type across the sampled sites. Another potential bias of our results is the insufficient representation of social insects. Ants and termites in our study were assessed using the same methods as for other macrofauna, but their distribution is very patchy. As mentioned above, especially the contribution of termites in deadwood, litter, and soil consumption in the tropics are likely to be underestimated in our soil food-web models[12,53]. Finally, knowledge on the feeding habits of soil animals is still fragmentary, especially in the tropics[43,46]. We based our food-web reconstruction on recent synthesis of feeding habits of soil animals, complemented with stable isotope analysis and other traits[43]. Still, many trophic links, such as feeding on fungi or bacteria, are difficult to quantify. To evaluate potential effects of this bias, we ran a sensitivity analysis for the degree of omnivory across food-web nodes. Despite considerable variation in the absolute estimates (Supplementary Fig. 6), we confirmed our main conclusions across a wide range of omnivory coefficients. In summary, we urgently call for studies on tropical soil animal feeding ecology to reduce uncertainty of estimates in soil food-web functioning.

Above, we summarized the structure and main energy pathways in soil food webs of temperate and tropical forests. Across a wide range of forest soil food webs, the highest energy fluxes were associated with litter consumption (an approximated average of 36%), followed by fungivory (21%), herbivory (14%), bacterivory (13%), soil consumption (8%), deadwood consumption (5%), and algivory (3%). However, we showed strong differences in soil food-web functioning across forest types. Temperate forests tended to the "brown" soil food web functional state, relying more on litter and microorganisms with low turnover (energy flux-to-biomass ratio) and predation. Tropical forests tended to the "green" state, relying more on freshly fixed plant biomass with high turnover and intensive predation. The similar trends in predation and herbivory point to a close association of predation with the "green" energy channel of soil food webs. High fungivory-to-bacterivory ratio in tropical soil food webs contrasts bacterial dominance in the bulk soil, emphasizing that food-web approaches are essential for the understanding of the ecosystem functioning and stability. We also observed a strong effect of earthworms on soil food web structure and trophic functions ("food-web engineering") in mixed broadleaved forests, where detrital pathways dominated. Finally, we showed that soil animals may process a high percentage of NPP in tropical ecosystems, highlighting the necessity to consider soil food webs in global biogeochemical models. Overall, our study demonstrated robust latitudinal patterns in soil food webs, which to a large extent are determined by the effect of temperature on animal metabolic rates, but may locally deviate due to dominance of keystone functional groups such as earthworms.

## Methods
### Field sites and sampling
The research presented here complies with all relevant ethical regulations. Collection of the Indonesian dataset was approved by the Indonesian Ministry of Forestry (PHKA, collection permit no. S.07/KKH-2/2013 and export permit no. 125/KKH-5/TRP/2014), Directorate General of Nature Resources and Ecosystem Conservation (KSDAE), and the Indonesian Institute of Sciences (LIPI, export permit no. 24/SI/MZB/IV/2014). The study represents a synthesis data analysis across research projects in Germany (beech forests)[54–56], European Russia

(southern taiga and mixed broadleaved forests)[34], Vietnam (monsoon forests), and Indonesia (rainforests)[46,57]. Parts of the data were published before (see references above), while the Vietnamese dataset is based on unpublished data. Selection of the ecosystems followed two main criteria: (1) representation of dominant forest types in temperate and tropical climates; (2) data on density and biomass collected with the same methods from the same sites across nematodes, meso- and macrofauna (litter and soil separated). In each region, 8 spatially distant (1–500 km apart) sampling sites were assessed, representing a specific forest type. Sites from the European Russia dataset were separated into two groups, 4 sites each, according to the two forest types (southern taiga vs mixed broadleaved forests; Supplementary Table 1).

Sampling was done in October 2014 in Germany, in October-November 2013 and 2016 in Indonesia (data from the two sampling years were averaged), in August 2018 in Russia and in June 2021 in Vietnam. The sampling timing reflects periods of high activity of soil invertebrates (optimal moisture and temperature conditions). We used the same set of methods across sites to collect soil animals. To sample nematodes, several soil cores of 5 cm diameter were taken at each study site: five cores were taken in rainforests and in beech forests, three in monsoon, and eighteen in mixed broadleaved and southern taiga forests. Each core comprised the full litter layer (OL and OF horizons) and the underlying soil to a depth of 5 cm (OH and part of the A horizon). Litter and soil layers were extracted separately using wet extraction with Baermann funnels. To sample meso- and macrofauna, we took several large cores for dry extraction: three 16 × 16 cm cores were taken in monsoon and rainforests, two 20 cm in diameter cores in beech, and six 20 × 20 cm cores (macrofauna) as well as eighteen 5 cm cores (mesofauna) in mixed broadleaved and southern taiga forests. Similar as for nematodes, each core comprised the litter and the underlying soil to a depth of 5 cm. Litter and soil samples were extracted separately using dry/heat extraction with Berlese funnels (Russia, Vietnam) or Kempson extractors[58] (Germany, Indonesia). All animals were extracted for 7–10 days with the extraction starting within a few days after sampling. Data for nematodes, meso- and macrofauna were averaged across all samples coming from the same plot within the same layer.

Environmental factors were assessed at each site in parallel to the soil invertebrate collections: soil pH was measured using a pH meter (KCl) from mixed soil samples (0–5 cm depth) and litter C/N ratio (as a proxy for litter quality) was measured using an elemental analyser from a representative mixture of freshly fallen senescent leaves. In addition, geospatial databases were mined for temperature and precipitation (CHELSA[59]), and NPP[60] (Supplementary Table 1). To compare NPP values with energy flux (see below), we converted data on energy flux from mW to $g\,m^{-2}\,y^{-1}$ (fresh biomass) roughly assuming 1 kg fresh biomass = $7 \times 10^6$ J[61].

### Identification and measurements
Meso- and macrofauna were identified under the dissecting microscope to order or family level and individual body lengths were measured. In case many individuals of the same group were present in the sample, we measured 10 random individuals to approximate mean body length. We then used taxon-specific body length–body mass regressions for microarthropods[62,63] and macrofauna[64] to calculate average fresh weight of each taxonomic group per site. For nematodes, 100 random individuals from each soil core were identified to genus level under the microscope and assigned to trophic groups of herbivores (H), bacterivores (B), fungivores (F), omnivores (O) and predators (P). The Nemaplex database[50] was used instead of direct measurements to calculate average body mass. All animals were divided into size classes based on equal intervals on a log10 body mass scale, independent of taxonomy. The following classes were distinguished: (1) 0.05–1.6 µg—nematodes B, F, H; (2) 1.6–50 µg—

springtails, mites, nematodes H, O, P, other microarthropods; (3) 0.05–1.6 mg—symphylans, pseudoscorpions, psocopterans, thrips, large mites, other small insects, spiders; (4) 1.6–50 mg—spiders, centipedes, beetles, woodlouse, small earthworms, orthoptera, other insects; (5) 0.05–1.6 g earthworms, opiliones, some other large arthropods.

## Food-web reconstruction

All data manipulations and statistical analyses were done in R v4.2.0 with R studio interface v2023.06.2 + 561 (RStudio, PBC). We used a "multichannel" soil food-web reconstruction approach including all three major size class compartments, i.e., soil micro-, meso- and macrofauna, modified by adding information derived from stable isotope data[16]. All consumers were classified into 32 trophic groups[65] represented by taxonomic groups in meso- and macrofauna (order or family level) and by the five trophic groups in nematodes (see above). The following parameters and traits were assigned to each trophic group: feeding preferences[43] to living plants (mainly sucking on roots and/or shoots), algae (microscopic algae, biofilms and lichens), bacteria, fungi, leaf litter, dead wood, soil organic matter or animal food (predation on other invertebrates), site-specific mean fresh body mass, body mass variation (we assumed standard deviation of 1 on a log10 body mass scale), site-specific group biomass per square meter, protection traits[43], and microhabitat preferences (soil, litter, ground, aboveground)[43]. The table was complemented with our data on average C and N percentages, $\delta^{15}N$ and $\delta^{13}C$ values (litter-calibrated) for each guild[34,46,66] (full data with all traits are available in Supplementary Data 1). We used the isotopic data to (1) refine preferences of all guilds to algae, as algivores typically have very low $\delta^{15}N$ values[67]—algivory was scaled from 0 to 100% using the approximate threshold of 2‰[67] and the minimum observed $\delta^{15}N$ of −1.4‰; (2) refine feeding preferences of earthworms to litter or soil organic matter, as endogeic species have higher $\delta^{15}N$ values[68] – feeding on soil was scaled from 0 to 100% using the observed minimum $\delta^{15}N$ of 1.2‰ and maximum of 5.5‰; (3) refine predation rates for omnivorous groups, as $\delta^{15}N$ values reflect trophic position in food webs[68,69]—auxiliary feeding on other invertebrates for omnivores was scaled from 0 to 100% using the observed minimum and maximum $\delta^{15}N$ in each food web; (4) calculate most probable predator-prey interactions based on isotopic distances among guilds and assuming trophic enrichment factors of 3.4 for $\delta^{15}N$ values and 0.4 for $\delta^{13}C$ values (see below).

Generic rules of food-web reconstruction based on food-web theory were used to infer weighted trophic interactions among all nodes with the following assumptions[16]: (1) There are phylogenetically inherited differences in feeding preferences for various basal resources and predation capability among soil animal taxa that define their feeding interactions (reflected as resource preferences, see above)[43]; (2) predator–prey interactions are primarily defined by the optimum predator-prey mass ratio (PPMR)[70]—typically, a predator is larger than its prey, but certain predator traits (hunting traits and behavior) can considerably modify the optimum PPMR[16]. We measured body mass distribution overlap for each potential pair of predator and prey in each food web to determine the most plausible trophic interactions. The optimum PPMR was set to 100[71]; (3) strength of the trophic interaction between predator and prey is defined by the overlap in their spatial niches related to vertical differentiation, with greater overlap leading to stronger interactions (i.e., no overlap among ground predators and endogeic earthworms); (4) predation is biomass-dependent[22]—due to higher encounter rate, predators will preferentially feed on prey that are locally abundant; (5) strength of the trophic interaction between predator and prey can be considerably reduced by prey protective traits—prey with physical, chemical or behavioral protection are consumed less[49]; (6) the closer are prey and predator in the isotopic space ($\delta^{15}N$ vs $\delta^{13}C$), accounting for trophic enrichment factors, the stronger is the feeding interaction among

them. All these assumptions were applied together to infer the most plausible trophic interaction matrix by multiplying all matrices, each scaled between 0 and 1. The respective R script is given in the Supplementary Data 1. Food-web reconstruction was carried out separately for each site. Sites were assumed to represent local food webs and were used as replicates in statistical analyses ($n$ = 32; Supplementary Figs. 1–4).

## Energy flux estimation

To calculate energy fluxes among food-web nodes, we used reconstructed interaction networks, biomasses, body mass-dependent metabolic losses, and environmental temperature, and applied the *fluxweb* package[22]. In brief, per-biomass metabolic rates were calculated from average fresh body masses using the equation and coefficients for corresponding phylogenetic groups of invertebrates[21]. The mean annual temperature was taken from public sources (rainforests– 25.2 °C, monsoon forests–26.0 °C, beech forests–7.5 °C, mixed broadleaved forests–7.1 °C, southern taiga forests–5.2 °C). The energy flux to each node was calculated from per-biomass metabolism, accounting for assimilation efficiencies (proportion of energy from food that is metabolized by the consumer) and losses to predation assuming a steady-state energetic system (i.e., energetic losses from each node are compensated by the lower trophic levels; e.g., if predators are present in the system there is enough prey to sustain them)[19,22]. We used diet-specific assimilation efficiencies, which we calculated from nitrogen content of each prey/basal resource node using a published equation[72]. Assimilation efficiencies for basal resources were calculated as 21% for plant material, 18% for leaf litter, 13% for soil organic matter, 96% for bacteria and 36% for fungi, and between 50 and 99% across animal groups. Then, we applied the *fluxing* function to the reconstructed interaction networks, which delivered energy flux estimations among all food-web nodes. Data were expressed in milliwatt per square meter (watt = joule second⁻¹).

## Statistical analyses

To test our hypotheses, at each site we calculated community biomass and total energy flux (in mW) per square meter of all soil invertebrates and individual size classes. We also classified energy fluxes according to "trophic functions" representing sums of outgoing fluxes from specific resources consumed: herbivory (living plants), algivory (algae and lichens), bacterivory (bacteria), fungivory (fungi), litter consumption (leaf litter), deadwood consumption (deadwood) and soil consumption (soil organic matter); predation represented a sum of outgoing fluxes from all invertebrate nodes.

To analyze the distribution of biomass across forest types, layers (litter vs soil), and size classes (hypotheses 1 and 2), we first ran a mixed-effect model testing the effect of these factors on log10-transformed invertebrate biomasses using the *lme4* package[73]. Chi-square, significance, and degrees of freedom were approximated using Wald Chi-square tests using the *car* package[74]. We added a random intercept term for the site to account for interdependence of size classes and layers from the same site. To test hypotheses 3 and 4, we ran linear models for each trophic function (see above) with climate type (temperate or tropical) and forest type nested in the climate type as predictors. Due to high skewness, the data were log10-transformed to ensure homogeneity in the residuals of the model. To test for significant differences between specific forest types, we applied the Tukey HSD test using the *agricolae* package[75].

Due to uncertainty in the resource preferences assigned to specific trophic guilds, especially to omnivores, we ran additional sensitivity analysis to evaluate the effect of this uncertainty. We re-ran our food-web reconstructions and energy flux calculations 10 times, varying the importance of additional resources ("omnivory parameter") from 0 to 1 with a step of 0.1. After each run, we calculated trophic functions to see how the differences among forest types are

affected by this parameter (Supplementary Fig. 6). To assess the bias associated with the snapshot seasonal sampling in monsoon forests, we used data on the seasonal variation of macrofauna populations in the same National park Cat-Tien in 2004–2005[28]. We calculated biomass of all macrofauna in each month as % of the biomass observed in June. This value varied from 33% in April to 182% in July and August and averaged around 90%.

To reveal correlations among environmental variables, soil food web structure and functioning, we applied non-metric multi-dimensional scaling using the *vegan* package[76] at each study site ($k = 3$). We used incoming energy fluxes to all food-web nodes as the input data to represent the energetic structure of food webs. Differences in the energetic structure among forest types were consequently tested with anosim using the *vegan* package. Environmental variables (temperature, precipitation, NPP, soil pH, and litter C/N ratio) and trophic functions (see above) were fitted onto the scaling and correlated with the energetic structure using *envfit*.

## Reporting summary
Further information on research design is available in the Nature Portfolio Reporting Summary linked to this article.

## Data availability
Raw data underlying results of the present paper are available from Supplementary Data 1 and Figshare https://doi.org/10.6084/m9.figshare.29341058.

## Code availability
Statistical R code underlying results of the present paper is available from Supplementary Data 1 and Figshare https://doi.org/10.6084/m9.figshare.29341058.

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

## Acknowledgements

The work was supported by the Alexander von Humboldt foundation in the framework of a Research group linkage programme 1071297—RUS—IP "Structure and functioning of belowground food webs across temperate and tropical ecosystems". A.M.P. was supported by the Deutsche Forschungsgemeinschaft (DFG, German Research Foundation) in the framework of the Emmy Noether program (Project number 493345801) and iDiv (DFG–FZT 118, 202548816). S.S. and V.K. were supported by DFG in the framework of the collaborative German–Indonesian research project CRC990 – EFForTS (192626868—SFB 990). M.M.P. and S.L.B. were funded by the DFG Priority Program 1374 "BiodiversityExploratories" (SCHE 376/38-2). A.K. was supported by state assignment of the Institute of Biology, Komi Scientific Centre, Ural Branch, Russian Academy of Sciences, no.125013101229-9.

## Author contributions

A.M.P., S.S., and A.V.T. conceptualized the idea and designed the study. I.S., S.L.B., V.K., A.K., V.M., M.M.P., O.R., S.M.T., A.G.Z., and A.I.Z. collected data. A.M.P. compiled the data, did data analysis and wrote the manuscript. Z.Z. contributed to statistical analysis and discussion of results. All authors contributed to the interpretation of the results and edited manuscript drafts.

## Funding

## Competing interests

The authors declare no competing interests.
