## [Transparent Peer Review file · Nature Communications]

Energy and biomass distribution in soil food webs of temperate and tropical forests

Corresponding Author: Professor Anton Potapov

Version 0:

Reviewer comments:

Reviewer #1

(Remarks to the Author)

This study examined energy fluxes to investigate common patterns in biomass and energy distribution among micro-, meso-, and macrofauna in forest ecosystems ranging from the southern taiga to tropical rainforests. The researchers discovered that tropical soil food webs exhibited higher predation rates compared to temperate soil food webs and relied more on plant consumption while relying less on bacterial, fungal, and litter consumption. Earthworms were identified as key players in promoting detrital energy pathways in mixed broadleaved forests, overriding climate-related differences across forests. The authors provided insights into energy and biomass distribution in soil food webs of temperate and tropical forests. However, before publication, I identified several apparent flaws and mistakes in the fieldwork and data analysis.

Firstly, yes, I have carefully reviewed your previous manuscript titled "Rainforest Transformation Reallocates Energy from Green to Brown Food Webs". The study examined changes in energy flow within aboveground and belowground animal food webs in tropical rainforests and plantations in Sumatra, Indonesia. The current manuscript investigated energy flow to analyze patterns in biomass and energy distribution across micro-, meso-, and macrofauna in forest ecosystems ranging from the southern taiga to tropical rainforests. I am curious about what specific improvements have been made to this manuscript, in addition to the comparison of different forest ecosystems. Regarding the forest ecosystems, why did the authors choose these four specific forest ecosystems when discussing forest ecosystems?

Secondly, I have observed that there are only 4 samples taken for the southern taiga and mixed broadleaved forests. Considering the wide range of taxonomic groups found in forest site, I have serious doubts about the accuracy of such a small sample size in representing the specific climatic region. It is highly unlikely that only four or eight samples can effectively capture the diversity and complexity of these forests. This limited sampling could potentially lead to the dominance of detrital energy pathways, masking any climate-related variations between the forests in your results. Additionally, the small number of soil samples may not provide a true reflection of the environment present at the sampling sites. As a result, the reliability of the soil sampling data is questionable.

(Remarks on code availability)

Reviewer #2

(Remarks to the Author)

This manuscript is a report of a field survey of soil invertebrates in a series of forest ecosystems along a climate gradient (temperate, tropical). Measurements include micro- (nematodes), meso- (springtails, mites, enchytraeids) and macrofauna (insects, myriapods, isopods, large arachnids, and earthworms). Herewith a quite complete picture of the soil food webs is obtained (without micro-organisms and protists). The choice of faunal groups captures most feeding trophic behaviours and diets, and the standardized methods make it possible to make straightforward comparisons. The study has produced a wealth of information regarding the structure and functionality of the forest soils food webs in association with climate. Herewith the results of the study are scientifically interesting and important and deserve publication in a high-quality peer-reviewed scientific journal.

However, the present manuscript is not (yet) suitable for publication in Nature Communications for the following reasons:

1. Introduction: I found the hypotheses difficult to read and to some extent not very well justified. For example, hypothesis 1 predicts that high metabolic costs lead to lower (equilibrium) biomasses. If metabolic costs are defined as a rate (?), then one should be careful to link such a rate to a state (equilibrium biomasses). Equilibrium biomasses may strongly depend on limiting factors, and on trophic interactions. For example, for the simplest predator-prey system, modelled with Lotka-Volterra equations, the equilibrium biomass/abundance of the predator is independent of its turn-over rate, while the equilibrium biomass of the prey does depend on the predator turn-over rate. This comment may also apply on what is said in line 383-387 where linkages between fluxes and equilibrium biomasses are mentioned. In hypothesis 2, it is stated that 'sublinear scaling of body size with metabolic rate (is that the same as metabolic costs in hypothesis 1?) and temperature, limits growth rate'. Such a statement might be true, but for some readers it might be helpful when this is somewhat more explained. In hypothesis 3, high predator-prey encounter rates are assumed to increase energy (biomass?) at the higher trophic levels. Again, I have difficulty to understand this. Finally, in hypothesis 4, animals with a higher energy demand should compete more strongly. However, competition may also depend on the biomasses of the competing species. On overall, the hypotheses may be correct, but for me they require some what more explanation and explicit definitions of the terms used.

2. In addition to my comment 1. it would have been helpful when a table with dimensions/units is given for all components and attributes of the food webs under study, such as body size, body mass, biomass, fluxes, metabolic rate, costs, expenses etc.

3. Energy is a key word in the manuscript, but I have difficulty is understanding what precisely is meant with energy. Food web fluxes are given in mW, but I could not figure out how these mW are calculated. I looked into the model by Gauzens et al. (ref 22). They also give fluxes in mW (in fact J.s⁻¹). But the equations, as far as I could see, calculate fluxes in terms of material flow rates. If so, they can be better expressed in terms of e.g. mg.s⁻¹. Expressing fluxes in mW suggest that they are (in a way) based on calorimetric/heat flow measurements. Maybe I have overlooked something, in the present manuscript or in the Gauzens et al. paper, but this might be something to be clarified.

4. Table 1 gives levels of statistical significance. These levels can also be given in the text. For a table like Table 1 prefer to see e.g. measured (average) values.

5. The figures are nicely designed, but in some cases, it is difficult to understand and interpret them. I would have liked (the addition) of some more simple figures. For example, bar diagrams showing total animal biomass for the various forest types. Or one showing the distribution of biomass over animal groups. In the present Figure 2 the reader is directed to differences in slopes, telling how biomass is distributed over different body mass classes and how this depends on forest type. Herewith one can understand what these slopes are, but less what the ecological meaning is of differences between slopes. In the discussion (line 354) it is said that most forests show a non-linear/double mode biomass spectrum. Again, what does this mean ecologically? Figure 3 gives a lot of information. Is it correct that the reader should in particular look at the positions of open and closed symbols to see differences in food web topology?

6. The discussion gives an elaborate overview of the results in the context of what is expected and what is in the literature. It is a solid and seemingly complete analysis of the results obtained, and that is good (although sometimes it is a little hard to see the forest through the trees). I see the difficulty to come up with generalisations. Has that something to do with 'idiosyncrasy', not uncommon in food webs? One more or less unifying pattern is that predation is 'associated' (?) the green (herbivory) energy channel. This seems a challenging results, as for many soil food webs it is thought that most energy comes from the brown (detritus) channel. Maybe this can be discussed in somewhat more detail.

I think some of my comments are due to the fact that the many results from this large field survey are presented in the form of a relatively short paper. This makes that the text is quite condense, making it sometimes hard for the reader to understand what is written. Some of my comments might also be due to overlooking something or lacking some understanding. My recommendation would be that it might be better to publish the results in a journal without a strict word limit. Then it becomes easier to explain the why, how, what, and so what of the results in a clear, comprehensive, and convincing way.

Hope my review is helpful.

(Remarks on code availability)

Version 1:

Reviewer comments:

Reviewer #1

(Remarks to the Author)

Thank you for your excellent revisions to the manuscript. I have another significant question about it. Seasonal sampling inconsistency fundamentally undermines cross-biome comparisons. Energy fluxes were measured during peak wet-season activity in tropical forests (Vietnam: June; Indonesia: Oct-Nov) versus late-season declines in temperate forests (Europe: October; Russia: Aug-Nov). This design artifactually inflates tropical fluxes—particularly monsoon forest estimates (e.g.,

1,090 vs. temperate 26–109 mW m⁻²)—while underestimating temperate systems where key decomposers (e.g., earthworms) peak in spring/summer. Critically, extrapolating wet-season fluxes to annual budgets yields biologically implausible claims like "81% of NPP consumed by monsoon forest soil fauna", which exceeds realistic carbon budgets and misrepresents ecosystem energetics. Without normalizing for seasonality (e.g., via temperature-adjusted activity curves) or acknowledging sampled periods as non-representative snapshots, reported biome differences reflect methodological asymmetry rather than true ecological divergence, invalidating the core claim of a global "brown-to-green functional shift."

(Remarks on code availability)

Reviewer #2

(Remarks to the Author)

The authors have addressed my comments satisfactorily.

There is however one element in their response that I did not entirely follow. That is the choice of the unit mW for the food web fluxes while biomasses are expressed in terms of mg.m⁻². When fluxes would be expressed in terms of mg.m⁻².d⁻¹, then for the reader it becomes possible to relate biomasses and fluxes, for example in terms of relative rates. To my opinion this would make the results more insightful. I agree that using mW makes the results easier to compare with some other studies, but I would not prefer 'convenience' in the discussion above 'clarity' of the results. I also did not understand why transforming mW to mass units will introduce additional uncertainty. As far as I could check, all estimates of metabolic costs go back to measurements of CO₂ emission or O₂ consumption. If there is some uncertainty in transformation, then to me it seems that this is already in the present mW values. Or do I overlook something? Finally, there are recent studies in which food web fluxes are estimated using direct calorimetry (heat) measurements, indicating differences in patterns in energy and material fluxes. (see e.g. van Bommel et al., SBB 2024). Very interesting!

But on overall, this is not reason to reject the paper.

Peter de Ruiter

(Remarks on code availability)

Version 2:

Reviewer comments:

Reviewer #1

(Remarks to the Author)

I thank the authors for their patient and detailed responses and for the considerable effort invested in revising the manuscript. I am satisfied with the clarifications and modifications made by the authors regarding my primary concern about "seasonal sampling inconsistency." The authors appropriately acknowledged that "snapshot" sampling is a common limitation in large-scale soil ecology studies, which is a fair assessment. More importantly, the authors have substantially improved the manuscript's rigor and the credibility of its conclusions through the following key revisions: 1) The explicit mention of sampling seasons in the introduction and figures, coupled with the inclusion of a clear disclaimer at the beginning of the discussion, responsibly alerts readers to potential biases in absolute values. 2) Providing an estimate of the seasonal bias for the Vietnamese monsoon forest site (citing historical data) was a strong addition. The calculation suggests that the June biomass is approximately 10% higher than the annual average, yet this bias is insufficient to explain the order-of-magnitude differences in energy fluxes between monsoon and temperate forests, effectively supporting the robustness of the core findings. 3) The seemingly high estimate of "81% of NPP consumed" was convincingly contextualized with field observations and recent literature, emphasizing the rapid energy turnover in tropical soil food webs. The subsequent toning down of the conclusion in the discussion to a more cautious statement is appropriate.

Furthermore, the authors' defense of the central conclusion—the "brown-to-green functional shift" (i.e., the change in the proportions of trophic functions)—is persuasive. As this is a relative measure, it is indeed less sensitive to absolute seasonal fluctuations than the total flux values. Therefore, this key pattern is likely robust despite the acknowledged sampling limitations.

In summary, the authors have adequately addressed my main concerns through candid discussion, additional data analysis, and careful textual revisions. The revised manuscript effectively argues for the validity of its core conclusions while acknowledging methodological constraints, thereby significantly enhancing its scientific quality. I believe the manuscript in its current form meets the standard for publication and recommend acceptance.

Dima Chen

(Remarks on code availability)

Reviewer #2

(Remarks to the Author)
I agree with the revision.

(Remarks on code availability)

REVIEWER COMMENTS

Reviewer #1 (Remarks to the Author):

This study examined energy fluxes to investigate common patterns in biomass and energy distribution among micro-, meso-, and macrofauna in forest ecosystems ranging from the southern taiga to tropical rainforests. The researchers discovered that tropical soil food webs exhibited higher predation rates compared to temperate soil food webs and relied more on plant consumption while relying less on bacterial, fungal, and litter consumption. Earthworms were identified as key players in promoting detrital energy pathways in mixed broadleaved forests, overriding climate-related differences across forests. The authors provided insights into energy and biomass distribution in soil food webs of temperate and tropical forests. However, before publication, I identified several apparent flaws and mistakes in the fieldwork and data analysis.

Firstly, yes, I have carefully reviewed your previous manuscript titled "Rainforest Transformation Reallocates Energy from Green to Brown Food Webs". The study examined changes in energy flow within aboveground and belowground animal food webs in tropical rainforests and plantations in Sumatra, Indonesia. The current manuscript investigated energy flow to analyze patterns in biomass and energy distribution across micro-, meso-, and macrofauna in forest ecosystems ranging from the southern taiga to tropical rainforests. I am curious about what specific improvements have been made to this manuscript, in addition to the comparison of different forest ecosystems. Regarding the forest ecosystems, why did the authors choose these four specific forest ecosystems when discussing forest ecosystems?

Response: We appreciate your comments to both previous and this manuscript. The two manuscripts use essentially the same approach (energy flux calculations) to analyze the functioning of food webs and their potential impacts on ecosystem functions. However, the design, the scope of the food web, and the research questions are different. The 8 rainforest sites used in this study (Sumatra) are the same as in our previous paper and this paper is cited in the methods. The improvements/differences in food-web modelling can be summarized as follows:

- 1) We analyzed forests of different biomes, i.e. temperate and tropical forests allowing novel insight into differences in soil food webs across large spatial scales and temperatures.
- 2) We focused on the belowground food-web compartment only and included nematodes as important microbivores in soil food webs.
- 3) We separated algivory from herbivory on vascular plants and feeding on dead wood from feeding on litter as feeding processes due to a more detailed assessment of feeding preferences.
- 4) We used site-specific data on litter-calibrated $\delta^{15}\text{N}$ and $\delta^{13}\text{C}$ values to refine feeding preferences (algivory, soil and litter feeding in earthworms, omnivory versus predation, and predator-prey interactions). Details are given in LL 585-596

The food-web reconstruction algorithm originates from an earlier review and now includes micro-, meso- and macrofauna (Potapov 2022; <https://doi.org/10.1111/brv.12857>). We modified the text to highlight changes/improvements (LL 572-574): “We used a ‘multichannel’ soil food-web reconstruction approach including all three major size class compartments, i.e. soil micro-, meso- and macrofauna, modified by adding information derived from stable isotope data¹⁶.”

Our aim in the present paper was to quantify the variation in the structure and functioning of soil food webs along major environmental gradients associated with the differences between (close to) natural temperate and natural tropical forests. Selection of forest ecosystems followed two main criteria. First, selected forest types (coniferous, broadleaved, monsoon, rainforests) represent dominant forest types in temperate and tropical climates. Second, we used data collected using the same methods and including assessment of density and biomass of microfauna (nematodes), meso- and macrofauna from the same sites. With this design we do not intend to be comprehensive in terms of ecosystem representation, but the approach allowed comparing major forest types across broad geographical and temperature gradients. We now added criteria for the forest selection in the text (LL 518-521): “Selection of the ecosystems followed two main criteria: (1) representation of dominant forest types in temperate and tropical climates; (2) data on density and biomass collected using the same methods from the same sites across micro- (nematodes), meso- and macrofauna, and considering both leaf litter and soil.”

Secondly, I have observed that there are only 4 samples taken for the southern taiga and mixed broadleaved forests. Considering the wide range of taxonomic groups found in forest site, I have serious doubts about the accuracy of such a small sample size in representing the specific climatic region. It is highly unlikely that only four or eight samples can effectively capture the diversity and complexity of these forests. This limited sampling could potentially lead to the dominance of detrital energy pathways, masking any climate-related variations between the forests in your results. Additionally, the small number of soil samples may not provide a true reflection of the environment present at the sampling sites. As a result, the reliability of the soil sampling data is questionable.

Response: We included 8 sites in each region (Indonesia, Vietnam, Germany, European Russia). The two forest types in European Russia were distinguished because of very different tree composition, soil fauna composition, and large geographical distance (~500 km apart; 4 sites each). The replicate n = 4 seems limited, but the data underlying these points are very extensive, covering well the local-scale environmental heterogeneity. The data from southern taiga and mixed broadleaved forests comes from the project describing size structure of soil food webs (<https://doi.org/10.1002/ecy.3421>). In this project, we collected 42 spatially-distinct samples (i.e., fauna communities) from each site to represent various taxa, resulting in processing over 60,000 individual specimens across 8 sites. A comparable effort has been done at the other locations (less samples but a higher sampling area and similar number of individual specimens). Therefore, while we couldn’t cover some of the large-scale variation, our points represent robust data records for the sites investigated. Even with 4 to 8 forests sampled per

forest type, we were able to see clear and statistically significant differences among forest types in different climates. This suggests a systematic, non-random pattern. We do not see any reasons why limited number of forests sampled could lead to a bias towards one or another trophic function in the soil food web (e.g., more detritivory or herbivory), or that this bias could be removed by increasing the sample size. We agree, however, that we did not capture the full diversity and complexity of each forest type. Thus, we added a note of caution in the discussion (LL 474-477): “Although we did an extensive sampling at each of the study sites (6-42 spatially distinct samples collected per site) we admit that we unlikely captured the entire diversity and complexity of each forest type across the sampled sites.”

Reviewer #2 (Remarks to the Author):

This manuscript is a report of a field survey of soil invertebrates in a series of forest ecosystems along a climate gradient (temperate, tropical). Measurements include micro- (nematodes), meso- (springtails, mites, enchytraeids) and macrofauna (insects, myriapods, isopods, large arachnids, and earthworms). Herewith a quite complete picture of the soil food webs is obtained (without micro-organisms and protists). The choice of faunal groups captures most feeding trophic behaviours and diets, and the standardized methods make it possible to make straightforward comparisons. The study has produced a wealth of information regarding the structure and functionality of the forest soils food webs in association with climate. Herewith the results of the study are scientifically interesting and important and deserve publication in a high-quality peer-reviewed scientific journal.

However, the present manuscript is not (yet) suitable for publication in Nature Communications for the following reasons:

1. Introduction: I found the hypotheses difficult to read and to some extent not very well justified. For example, hypothesis 1 predicts that high metabolic costs lead to lower (equilibrium) biomasses. If metabolic costs are defined as a rate (?), then one should be careful to link such a rate to a state (equilibrium biomasses). Equilibrium biomasses may strongly depend on limiting factors, and on trophic interactions. For example, for the simplest predator-prey system, modelled with Lotka-Volterra equations, the equilibrium biomass/abundance of the predator is independent of its turn-over rate, while the equilibrium biomass of the prey does depend on the predator turn-over rate. This comment may also apply on what is said in line 383-387 where linkages between fluxes and equilibrium biomasses are mentioned. In hypothesis 2, it is stated that ‘sublinear scaling of body size with metabolic rate (is that the same as metabolic costs in hypothesis 1?) and temperature, limits growth rate’. Such a statement might be true, but for some readers it might be helpful when this is somewhat more explained. In hypothesis 3, high predator-prey encounter rates are assumed to increase energy (biomass?) at the higher trophic levels. Again, I have difficulty to understand this. Finally, in hypothesis 4, animals with a higher energy demand should compete more strongly. However, competition may also depend on the biomasses of the competing species. On overall, the hypotheses may be correct, but for me they require some what more explanation and explicit definitions of the terms used.

Response: We appreciate the critical assessment of our hypotheses and justified them better in the new version of the manuscript. Indeed, standing biomasses and process rate should be put in environmental and top-down context. We expected that standing biomass of invertebrate communities decreases towards the tropics because metabolic losses of consumers increase with temperature at a higher rate than the ecosystem productivity. We suggest that animals in the tropics consume a surplus of plant production due to their increased metabolic demands and that they are resource-limited. We also consider that increased predation rates may contribute to a lower biomass equilibrium in the tropics. In temperate ecosystems, animals are under little top-down control, but resource limitation can be driven by environmental factors such as low temperature and thus slow decomposition rates. We updated the text in the introduction (with references; LL 75-83): “Due to high temperature in tropical ecosystems, the metabolic rate (i.e. energy needed to sustain functioning of the organism) of organisms is high, necessitating high consumption rates. Ecosystem production increases with temperature at a lower rate than metabolic losses of consumers, leading to energy limitation, high predation rates, competition for resources and prey among soil animals, and potential changes in the use of basal resources and related functioning in soil food webs of tropical forests. Testing these theoretical predictions would allow us to apply food-web approaches in cross-ecosystem models of soil organic matter dynamics and to better understand the factors driving terrestrial biodiversity”. We also updated our hypothesis 1: “Tropical soil invertebrate communities have lower total biomass, but larger energy use per unit of time than temperate ones because of higher resource limitation and predation under high temperatures.” We now mention the environmental constraints in the discussion (LL 397-399): “Due to environmental limitations of decomposition rate, litter accumulates in many temperate forests and hosts large microbial biomass stocks, thereby providing ample living space and resources for soil animals.” And LL 296-298: “We hypothesized that tropical soil food webs cannot sustain large community biomass because of higher metabolic rate and increased resource limitation and predation.”

Hypothesis 2 is now specified and explained better (LL 96-98): “Proportionally more energy is processed by small-sized animals in the tropics because temperature poses limitations on growth and body mass, and more energy is spent for respiration.”

We assume that the energy flux to predators (which is a proxy of prey biomass consumption) is driven by the effectiveness of the predators, i.e. how much energy they have to spend for searching/capturing the prey. We specified Hypothesis 3 (LL 98-101): “Proportionally more energy is propagating to higher trophic levels in the tropics due to shallow litter layer and high animal activity resulting in lower search time and higher predator-prey encounter rate (i.e., predators spend less energy for prey searching and capturing).” Increase in predation rates with temperature is commonly observed in laboratory experiments, which we mention in the discussion (<https://royalsocietypublishing.org/doi/10.1098/rspb.2016.2570>, <http://www.nature.com/articles/s41558-017-0002-z>).

Hypothesis 4 was specified and explained better (LL 101-103): “Due to high temperature, rapid decomposition (shallow litter layer), high energy demand and therefore high competition for

resources in the tropics, consumption of easily accessible resources with fast turnover is promoted...”

2. In addition to my comment 1. it would have been helpful when a table with dimensions/units is given for all components and attributes of the food webs under study, such as body size, body mass, biomass, fluxes, metabolic rate, costs, expenses etc.

Response: The attributes/variables are introduced one by one in the results and thus we decided to incorporate definitions as they appear in the text. To make the terminology clearer, the terms ‘metabolic costs’ and ‘metabolic expenses’ were removed from the manuscript as redundant.

L 41 ‘Energy flux in the ecosystem, i.e. the total amount of energy transferred in food webs, is associated with both ecosystem biodiversity and functioning’

L 76 ‘Due to high temperature in tropical ecosystems, the metabolic rate (i.e., the energy needed to sustain functioning of the organism)...’

L 111 ‘Soil animal biomass (i.e., the sum of individual masses in a community)’

L 138 ‘There was a gradual increase in the biomass – individual body mass slope (i.e. increase in large-sized animals)’

3. Energy is a key word in the manuscript, but I have difficulty in understanding what precisely is meant with energy. Food web fluxes are given in mW, but I could not figure out how these mW are calculated. I looked into the model by Gauzens et al. (ref 22). They also give fluxes in mW (in fact J.s⁻¹). But the equations, as far as I could see, calculate fluxes in terms of material flow rates. If so, they can be better expressed in terms of e.g. mg.s⁻¹. Expressing fluxes in mW suggest that they are (in a way) based on calorimetric/heat flow measurements. Maybe I have overlooked something, in the present manuscript or in the Gauzens et al. paper, but this might be something to be clarified.

Response: Energy fluxes were calculated based on per-biomass metabolism. We agree that Watts could be transformed to their mass equivalents and this is often done in ecosystem modelling. However, we prefer to keep watts for two reasons: (1) watts are now routinely used in other energy flux papers as SI units and this makes our study easier to compare to other studies (e.g. <https://www.nature.com/articles/s41467-024-54401-z>, <https://onlinelibrary.wiley.com/doi/10.1002/ece3.8060>, <https://onlinelibrary.wiley.com/doi/10.1111/gcb.17554>); (2) watts (joules per second) are used in metabolic rate equations and energy flux calculations. Transforming them to mass units will introduce additional uncertainty as the energy equivalent of (bio)mass may vary depending on the composition of this biomass.

4. Table 1 gives levels of statistical significance. These levels can also be given in the text. For a table like Table 1 prefer to see e.g. measured (average) values.

Response: We now transferred statistical outputs to the text and give mean values in Table 1 instead.

5. The figures are nicely designed, but in some cases, it is difficult to understand and interpret them. I would have liked (the addition) of some more simple figures. For example, bar diagrams showing total animal biomass for the various forest types. Or one showing the distribution of biomass over animal groups. In the present Figure 2 the reader is directed to differences in slopes, telling how biomass is distributed over different body mass classes and how this depends on forest type. Herewith one can understand what these slopes are, but less what the ecological meaning is of differences between slopes. In the discussion (line 354) it is said that most forests show a non-linear/double mode biomass spectrum. Again, what does this mean ecologically? Figure 3 gives a lot of information. Is it correct that the reader should in particular look at the positions of open and closed symbols to see differences in food web topology?

Response: Thank you. We prefer to keep Figures in the present form, however, we added explanations to the text and Table 1 is updated. Diagrams showing total animal biomass (barplot) and biomasses of individual groups (bubble plot) are shown in Figure 1. The updated Table 1 gives additional easily interpretable summary of these results.

To clarify the ecological meaning of the slopes in Figure 2 we added explanations to the text (LL 138-139): “There was a gradual increase in the biomass – individual body mass slope (i.e. increase in large-sized animals) from the taiga to the tropics...” (LL 142-144): “The energy flux – body mass slope was slightly positive and similar in monsoon, taiga and mixed broadleaved forest (i.e. more energy is flowing to large-sized animals)...”. The ecological meaning of the bimodal body mass distribution is now explained in more detail in the discussion (LL 366-369): “In most forests we observed non-linear body mass spectra with a double mode distribution. Specifically, biomass and energy concentrated in ‘large microfauna and microarthropods’ and ‘macrofauna’ body mass classes, whereas ‘microfauna’ and ‘small macrofauna’ had low biomass and energy flux in comparison to other body mass classes.” Discussions on its ecological implications are given in the same paragraph.

We do not expect readers to inspect every food-web node and trophic interaction in Figure 3. Rather, this figure aims at summarizing our food-web reconstructions and illustrating this methodological step for transparency and validation of our approach. The presented food-web reconstructions are subsequently used for energy flux analysis which is presented in Figure 4. Visual comparison of the networks and dominant animal groups can easily be grasped by the reader. We now clarify the purpose of this figure in the text (LL 166-168): “Reconstructed soil animal trophic networks had more trophic levels and allocated more energy to macrofauna in the tropics than in temperate forests (networks are shown in Fig. 3 for illustration of the methodology and visual transparency of our results).”

6. The discussion gives an elaborate overview of the results in the context of what is expected and what is in the literature. It is a solid and seemingly complete analysis of the results

obtained, and that is good (although sometimes it is a little hard to see the forest through the trees). I see the difficulty to come up with generalisations. Has that something to do with 'idiosyncrasy', not uncommon in food webs? One more or less unifying pattern is that predation is 'associated' (?) the green (herbivory) energy channel. This seems a challenging results, as for many soil food webs it is thought that most energy comes from the brown (detritus) channel. Maybe this can be discussed in somewhat more detail.

Response: Thank you. Indeed, our discussion covers multiple topics and thus the main 'story line' is not always evident. This is explained by the multiple novel aspects associated with the comparison of tropical and temperate soil food webs. We summarise our main findings in the last paragraph of the discussion, which we now modified to include the association of predation with the green energy channel (LL 491-510):

“Above we summarised the structure and main energy pathways in soil food webs of temperate and tropical forests. Across a wide range of forest soil food webs, the highest energy fluxes were associated with litter consumption (an approximated average of 36%), followed by fungivory (21%), herbivory (14%), bacterivory (13%), soil consumption (8%), deadwood consumption (5%) and algivory (3%). However, we showed strong differences in soil food-web functioning across forest types. Temperate forests tended to the “brown” soil food web functional state, relying more on litter and microorganisms with low turnover (energy flux-to-biomass ratio) and predation. Tropical forests tended to the “green” state, relying more on freshly-fixed plant biomass with high turnover and intensive predation. The similar trends in predation and herbivory point to close association of predation with the “green” energy channel of soil food webs. A high fungivory-to-bacterivory ratio in tropical soil food webs contrasts bacterial dominance in bulk soil, emphasizing that food-web approaches are essential for the understanding of ecosystem functioning and stability. We also observed a strong effect of earthworms on soil food web structure and trophic functions (“food-web engineering”) in mixed broadleaved forests, where detrital pathways dominated. Finally, we showed that soil animals may process a high percentage of NPP in tropical ecosystems, highlighting the necessity to consider soil food webs in global biogeochemical models. Overall, our study demonstrated robust latitudinal patterns in soil food webs, which to a large extent are determined by the effect of temperature on animal metabolic rates, but may locally deviate due to dominance of keystone functional groups such as earthworms.”

I think some of my comments are due to the fact that the many results from this large field survey are presented in the form of a relatively short paper. This makes that the text is quite condense, making it sometimes hard for the reader to understand what is written. Some of my comments might also be due to overlooking something or lacking some understanding. My recommendation would be that it might be better to publish the results in a journal without a strict word limit. Then it becomes easier to explain the why, how, what, and so what of the results in a clear, comprehensive, and convincing way.

Hope my review is helpful.

Response: We appreciate all the comments and suggestions for improvement. We hope that with additional explanations our text is now more comprehensible and all points are well explained.

Reviewer #1 (Remarks to the Author):

Thank you for your excellent revisions to the manuscript. I have another significant question about it. Seasonal sampling inconsistency fundamentally undermines cross-biome comparisons. Energy fluxes were measured during peak wet-season activity in tropical forests (Vietnam: June; Indonesia: Oct-Nov) versus late-season declines in temperate forests (Europe: October; Russia: Aug-Nov). This design artifactually inflates tropical fluxes—particularly monsoon forest estimates (e.g., 1,090 vs. temperate 26–109 mW m⁻²)—while underestimating temperate systems where key decomposers (e.g., earthworms) peak in spring/summer. Critically, extrapolating wet-season fluxes to annual budgets yields biologically implausible claims like "81% of NPP consumed by monsoon forest soil fauna", which exceeds realistic carbon budgets and misrepresents ecosystem energetics. Without normalizing for seasonality (e.g., via temperature-adjusted activity curves) or acknowledging sampled periods as non-representative snapshots, reported biome differences reflect methodological asymmetry rather than true ecological divergence, invalidating the core claim of a global "brown-to-green functional shift.

Reply: We agree that the snapshot sampling introduces biases in the comparison across climate types. Unfortunately, this is a general problem for most global-scale analyses of soil biodiversity (e.g. van den Hoogen et al. 2019; Phillips et al. 2019). Laborious data collection leads to a lack of time series data, which is one of the most pressing knowledge gaps in modern soil macroecology that requires cross-country efforts of large consortia to address (e.g. Potapov et al. 2022; Ganault et al. 2024; Mathieu et al. 2024). This knowledge gap is even more evident in tropical ecosystems. To navigate potential misreading of our paper, we now introduced the following changes:

1) We highlighted seasons of sampling in each of the forest type in the end of the introduction and Figure 1 to make it clear that snapshots are compared. In the introduction (LL 88-92): "To achieve this, we collected data on density and body size of soil nematodes, microarthropods and macrofauna from 32 sites across four regions – Germany, European Russia, Indonesia and Vietnam in the periods of high activity of soil invertebrates (late summer or autumn in the temperate climate, early wet season in the tropical climate)." Furthermore, we added a disclaimer in the beginning of discussion (LL 284-286): "Our comparison is based on snapshot data of peak soil fauna activities, so the absolute values should be treated with caution. Nevertheless, the proportional changes of different trophic functions are likely representative for the climate types."

2) We estimated the potential bias in the Vietnam data using data of our colleague on the seasonal variation of macrofauna populations in the same National Park Cat-Tien in 2004-2005 (Figure R1 below). In our food-web model, the seasonal variation in metabolic activity of fauna related to seasonal moisture variation could be reflected using the seasonal changes in the population density of different functional groups of soil fauna. To estimate the bias, we calculated probable biomass of all macrofauna in each month as % of the biomass observed in June. This value varied from 33% in April

to 182% in July and August. On average across the year, this value was 90%, meaning that the biomass of macrofauna observed in June in our assessment is about 10% higher than the average biomass of macrofauna throughout the year in our sampling sites. Since macrofauna dominates in monsoon forests (Figures 1, 2 and 3 of the main manuscript), these variations in macrofauna biomass should largely correlate to the total energy flux of the soil food web. Even unexpectedly high densities in our snapshot assessment couldn't explain the order-of-magnitude difference in energy fluxes between monsoon and temperate forests. Thus, our conclusions are robust despite potential bias in absolute values. Unfortunately, we cannot publish detailed raw data presented in the Figures below as owner of these data left academic science ca. 10 years ago. These data summaries were published in Russian as a book chapter (Anichkin 2011) and in an PhD thesis (Anichkin 2008). We now briefly presented these calculations in the discussion (LL 333-339) and methods (LL 685-689).

Figure R1. Seasonal variation in abundance (bars) and fresh biomass (points connected with black line) of litter and soil-dwelling macrofauna from Cat-Tien National Park in Vietnam collected using hand sorting. The sampling was done in 2004-2005 in September (S), November (N), January (J), April (A), June (J), July (J), and November (N; order the same as in the x axis labels). Means and standard errors are shown. Data from four sites with mixed monsoon forest vegetation within the same sampling area

(~25 km²) as in the main study analysis are shown (dominant tree species: *Azadirachta* sp., *Dipterocarpus* sp., *Lagerstroemia calyculata*). Figures are based on (Anichkin 2011) and unpublished PhD thesis (Anichkin 2008).

Despite consumption of 81% of NPP looks unrealistic, personal observations of the authors in the field sites in Vietnam suggest that nearly all leaf litter accumulated during the dry season in which litterfall occurs disappears within a few weeks or even days after the first rains due to the high activity of millipedes, termites and other large detritivores. This aligns with recently published estimates of litter consumption by soil fauna (Heděnc et al. 2022). Approximately 80% of consumed litter returns to the system in the form of unassimilated material (faeces) and further enters the nutrient and energy cycles in the soil or in termite mounds. Thus, we don't agree that this number exceeds realistic carbon budgets and misrepresents ecosystem energetics. It can rather be argued that the carbon budgets and ecosystem energetics may need to be revisited. We added the information above to the discussion and edited conclusions of this paragraph as following: "While keeping in mind the uncertainty of absolute values, we still can conclude that tropical animal soil food webs channel a much larger part of the NPP than temperate ones and this coincides with lower accumulation of topsoil organic matter in the former" (LL 345-354).

Summer months in Europe are dry and peak activities of soil fauna occur in late spring and autumn. Therefore, our sampling represents peak activity in temperate forests, rather than a late season decline. We also disagree that uncertainty associated with "seasonal-snapshot" sampling undermines our claim of the temperate-to-tropical brown-to-green functional shift as this ratio is independent from the total energy flux values and represents the % detritivory against % herbivory. Detritivory in temperate climates might change throughout the year, but it is unlikely to peak in August (Russian dataset) or early October (German dataset), when the litterfall is about to start, while the leaves from the last year are heavily decomposed. Our sampled communities should represent the food resources in the soil food web of the last months *before* the sampling (i.e. summer, early autumn). According to previous research, there is no evidence of seasonal % detritivory shifts in large decomposers such as millipedes, while such changes in springtails are forest type-specific (Potapov et al. 2014). We now highlight this debate as a research question to address in future studies: LL 430-431.

van den Hoogen, J.; Geisen, S.; et al., Crowther, T. Soil Nematode Abundance and Functional Group Composition at a Global Scale. *Nature* **2019**, *572*, 194–198. <https://doi.org/10.1038/s41586-019-1418-6>.

Phillips, H. R. P.; et al. Eisenhauer, N. Global Distribution of Earthworm Diversity. *Science* **2019**, *366* (6464), 480–485. <https://doi.org/10.1126/science.aax4851>.

Potapov, A. M. et al. Wall, D. Global Monitoring of Soil Animal Communities Using a Common Methodology. *Soil Organisms* **2022**, *94* (1), 55–68. <https://doi.org/10.25674/so94iss1id178>.

Ganault, P.; et al. Eisenhauer, N. Soil BON Earthworm - A Global Initiative on Earthworm Distribution, Traits, and Spatiotemporal Diversity Patterns. *Soil Organisms* **2024**, *96*, 47–60. <https://doi.org/10.25674/362>.

Mathieu, J. et al. Eisenhauer, N. sOilFauna - a Global Synthesis Effort on the Drivers of Soil Macrofauna Communities and Functioning: WORKSHOP REPORT. *Soil Organisms* **2022**, *94* (2), 111–126. <https://doi.org/10.25674/SO94ISS2ID282>.

Anichkin, A. E. (2008) Structure and functional significance of soil animal communities in a monsoon tropical forest. Unpublished PhD thesis, Moscow, IEE RAS (in Russian).

Anichkin, A. E. (2011) Soil macrofauna: structure and seasonal dynamics. In Tiunov, A. V. (ed.) Structure and functions of soil communities of a monsoon tropical forest (Cat Tien National Park, southern Vietnam). Moscow, KMK Scientific Press (in Russian).

Heděnc, P.; Jiménez, J. J.; Moradi, J.; Domene, X.; Hackenberger, D.; Barot, S.; Frossard, A.; Oktaba, L.; Filser, J.; Kindlmann, P.; Frouz, J. Global Distribution of Soil Fauna Functional Groups and Their Estimated Litter Consumption across Biomes. *Sci Rep* **2022**, *12* (1), 17362. <https://doi.org/10.1038/s41598-022-21563-z>.

Potapov, A. M.; Semenyuk, I. I.; Tiunov, A. V. Seasonal and Age-Related Changes in the Stable Isotope Composition ($^{15}\text{N}/^{14}\text{N}$ and $^{13}\text{C}/^{12}\text{C}$) of Millipedes and Collembolans in a Temperate Forest Soil. *Pedobiologia* **2014**, *57* (4–6), 215–222. <https://doi.org/10.1016/j.pedobi.2014.09.005>.

Reviewer #2 (Remarks to the Author):

The authors have addressed my comments satisfactorily.

There is however one element in their response that I did not entirely follow. That is the choice of the unit mW for the food web fluxes while biomasses are expressed in terms of mg.m⁻². When fluxes would be expressed in terms of mg.m⁻².d⁻¹, then for the reader it becomes possible to relate biomasses and fluxes, for example in terms of relative rates. To my opinion this would make the results more insightful. I agree that using mW makes the results easier to compare with some other studies, but I would not prefer 'convenience' in the discussion above 'clarity' of the results. I also did not understand why transforming mW to mass units will introduce additional uncertainty. As far as I could check, all estimates of metabolic costs go back to measurements of CO₂ emission or O₂ consumption. If there is some uncertainty in transformation, then to me it seems that this is already in the present mW values. Or do I overlook something? Finally, there are recent studies in which food web fluxes are estimated using direct calorimetry (heat) measurements, indication differences in patterns in energy and material fluxes. (see e.g. van Bommel et al., SBB 2024). Very interesting!

But on overall, this is not reason to reject the paper.

Peter de Ruiter

Reply: We agree that there are arguments for both Watts and grams. To keep both convenience and clarity, we now additionally reported fluxes in $\text{g m}^{-2} \text{y}^{-1}$ for total fluxes per ecosystem type in Figure 3 and presented NPP values in $\text{g m}^{-2} \text{y}^{-1}$ in the text.

Thank you for recommending the interesting paper. We will look into this topic more in detail.